# Effects of Waste Expanded Polypropylene as Recycled Matrix on the Flexural, Impact, and Heat Deflection Temperature Properties of Kenaf Fiber/Polypropylene Composites

**DOI:** 10.3390/polym12112578

**Published:** 2020-11-02

**Authors:** Junghoon Kim, Donghwan Cho

**Affiliations:** Department of Polymer Science and Engineering, Kumoh National Institute of Technology, Gumi, Gyeongbuk 39177, Korea; kjh40210@naver.com

**Keywords:** waste expanded polypropylene, natural fiber composites, recycling, properties, injection molding

## Abstract

Waste Expanded polypropylene (EPP) was utilized as recycled matrix for kenaf fiber-reinforced polypropylene (PP) composites produced using chopped kenaf fibers and crushed EPP waste. The flexural properties, impact strength, and heat deflection temperature (HDT) of kenaf fiber/PP composites were highly enhanced by using waste EPP, compared to those by using virgin PP. The flexural modulus and strength of the composites with waste EPP were 98% and 55% higher than those with virgin PP at the same kenaf contents, respectively. The Izod impact strength and HDT were 31% and 12% higher with waste EPP than with virgin PP, respectively. The present study indicates that waste EPP would be feasible as recycled matrix for replacing conventional PP matrix in natural fiber composites.

## 1. Introduction

Environmental issues due to rapidly increasing waste plastics have been importantly dealt with not only in advanced countries, but also in developing countries. Many researches have been performed to reduce or solve the environmental problems due to industrial waste plastics [1,2,3]. A huge amount of commercial plastics including polyolefin-based products has been utilized and discarded in our daily life. Some of them can be collected for recycling or reuse.

Industrial waste thermoplastics can be well matched with plant-based natural fibers such as kenaf, jute, flax, hemp, etc. to make natural fiber composites, also referred to as biocomposites, with enhanced properties. Natural fiber composites have several advantages such as acceptable mechanical properties, light weight, low cost, environmental friendliness, carbon dioxide reduction, etc. over conventional glass fiber composites [4,5]. Hence, plant-based natural fibers have been widely used in producing natural fiber composites not only with thermoplastic polymers but also with thermosetting polymers [6,7,8].

Kenaf (*Hibiscus cannabinus*) is cultivated mainly in countries with subtropical climate. It is composed of approximately 45–57% cellulose, 21% hemicellulose, 8–13% lignin, 4% pectin, etc. [9]. It has merits such as fast growing, high carbon dioxide absorption during cultivation, relatively high mechanical, impact, and thermal properties, compared to other natural fibers. Hence, kenaf is one of the most frequently used fiber reinforcements for natural fiber composites [10,11,12]. One of the most popular thermoplastic resins used in the composites with kenaf fibers is polypropylene [13,14,15].

Polypropylene (PP) is one of the most frequently used general-purpose thermoplastics in many industrial applications. It can often be produced via extrusion and injection molding processes not only with glass and carbon fibers, but also with natural fibers. Several papers have dealt with PP and kenaf fiber to make the composite. Most recently, Nematollahi et al. reported that composites with neat PP and 20 wt% kenaf fiber were fabricated by using extrusion injection molding and carried out experimental and numerical studies of the critical fiber length to affect the load transfer efficiency and stiffening of resulting composites [16]. They also studied that morphology, thermal, and mechanical properties of extruded injection molded kenaf fiber reinforced PP composites [17]. Radzuan et al. investigated focusing on machinability and moldability of kenaf composites with PP, PLA, and epoxy for automotive components, which was produced by injection and compression molding processes [18]. Islam et al. characterized the effect of alkali-treatment on the interfacial and mechanical properties of kenaf/recycled PP composites made by using extrusion and injection molding methods [19]. However, kenaf fiber composites with waste expanded polypropylene (EPP) have been rarely found.

EPP can be mainly manufactured by foaming process with PP beads, as described elsewhere [20,21]. EPP foams exhibit excellent impact resistance, insulation, energy absorption, dimensional stability, etc. [22]. Hence, they are often used as foam materials for insulation, protection, and packaging in many industrial and personal applications [23,24]. A large amount of expanded plastics has been discarded as waste after end-used or landfilled [25]. Chemical and thermal processes to recycle waste plastics are well-known but they are costly and difficult to use [26]. One of the possible and simple approaches to solve such the problems is to develop composite materials with biomass-based reinforcements and waste plastics together [27,28]. Although there are a few papers [29] reporting on the extrusion process of expanded polystyrene with biomass, studies on producing natural fiber composites using waste EPP only by injection molding process in the absence of extrusion process have been scarcely found.

It has been well known that thermoplastic resins have advantages such as no cure reaction, recyclability, fast processing time, and good fracture toughness over thermosetting resins, whereas they also have processing difficulties due to high melt viscosity and low resin flow, causing inefficient resin impregnation. For this reason, extrusion and injection molding processes have been most frequently used processing methods for producing fiber-reinforced thermoplastics with chopped fibers. However, these processes often result in shortening of reinforcing fiber length, to be less than a few hundred micrometers. The mechanical and impact properties of resulting composites are much lower than expected. It restricts their extensive applications. Uses of increased fiber loadings may make thermoplastic composite processing more difficult because shear forces occurring between the screws in the barrel are increased during melt compounding process, resulting in further fiber damages and shortening [30]. It may be expected that performing injection molding by directly feeding the fiber and resin without an extrusion stage provides benefits making composite processing simpler.

Consequently, the objective of this study is to diagnose the feasibility of using waste EPP as recycled matrix for environmentally benign natural fiber composites. For this, novel composites consisting of chopped kenaf fibers and waste EPP were produced only by using injection molding process without extrusion process. The effect of waste EPP on the flexural, impact, and heat deflection temperature properties of kenaf fiber composites consisting of the PP matrix derived from melted waste EPP was investigated, comparing to those of conventional kenaf fiber composites consisting of virgin PP matrix.

## 2. Materials and Methods

### 2.1. Materials

In the present work, kenaf fibers, which were cultivated, extracted, and supplied from Bangladesh Jute Research Institute (BJRI), Bangladesh, were used as reinforcement. ‘As-supplied’ kenaf fibers were of bundle form of 70–80 mm long. The kenaf bundles were chopped to about 4–5 mm long by using a pulverizing machine (DHS-28, Man Pyung Co., Daegu, Korea). Virgin PP beads of roughly 3 mm diameter (SEETEC R3410, LG Chemical Co., Seoul, Korea) were used to produce commercial EPP foams. The PP bead has the density of 0.9 g/cm^3^ and the melt flow index of 7 g/10 min. According to the manufacturer’s information, the virgin PP and the PP used to produce commercial EPP were identical with the same molecular weight. Waste EPP, that is, industrial EPP blocks discarded after end-used, or due to defects and damages, were kindly supplied from KOSPA Co., Korea. The dimensions of waste EPP blocks of a rectangular shape were 500 mm × 300 mm × 60 mm. The waste EPP blocks were cut to smaller rectangular shapes by using a band saw (BAS 250 G, ELEKTRA BECKUM GmbH, Meppen, Germany) and then crushed to about 4–5 mm long by using a pulverizing machine (DHS-28, Man Pyung Co., Daegu, Korea). Here, the word ‘waste EPP’ indicates the EPP obtained by crushing process.

### 2.2. Processing of Kenaf Fiber/PP Composites through Recycling of Waste EPP

Kenaf fiber/PP composites were produced only by using injection molding process in the absence of extrusion process. Figure 1 displays crushed waste EPP and chopped kenaf fibers obtained by pulverizing waste EPP blocks and kenaf fiber bundles, prior to injection molding process to make the composites. Crushed waste EPP and chopped kenaf fibers were sufficiently dried at 70 °C for 6 h in a convection oven.

They were uniformly mixed by a manual manner and then regularly fed into the hopper during the injection process. The kenaf fiber contents were 10, 20, and 30% by weight. Injection molding process was carried out by using an injection machine (PRO-WD 80, Dong Shin Co., Changwon, Korea). The barrel temperature was increased stepwise by 5 °C from 155 °C in the entering zone to 170 °C in the nozzle zone. The mold temperature was 80 °C. The holding pressure was 20 kg/cm^2^ and the holding time was about 3 s. The injection pressure was 25–35 kg/cm^2^ and the injection time was about 2 s. During injection molding, the crushed waste EPP was transformed to the PP matrix surrounding individual kenaf fibers by complete melting. The specimens for flexural, impact, and heat deflection temperature tests of kenaf fiber/PP composites with varying the kenaf fiber contents were directly obtained from injection molding process.

For comparison, the specimens made with the virgin PP beads, the specimens made with the crushed waste EPP without kenaf fibers, and the composite specimens made with the chopped kenaf fiber and the PP were also prepared by injection molding, respectively. Figure 2 shows the injection molding process for producing kenaf fiber/PP composites by using the virgin PP beads and the crushed waste EPP with varying the kenaf fiber contents, respectively.

### 2.3. Characterization of the Composites

#### 2.3.1. Microscopic Observation

Scanning electron microscopy (SEM, JSM-6380, JEOL Ltd., Tokyo, Japan) was used to observe the fracture surfaces of the composites. All the specimens were coated with platinum for 3 min by a sputtering method prior to observations. The acceleration voltage was 10 kV and the SEM images were obtained using a secondary electron image mode.

#### 2.3.2. Flexural Test

Three-point flexural tests were carried out according to the ASTM D790M standard using a universal testing machine (AG-50kNX, Shimadzu Co., Kyoto, Japan). The span-to-depth ratio of each rectangular specimen was 32:1. The load cell of 50 kN was used. The crosshead speed was 5.1 mm/min. Ten specimens were used to obtain the average flexural modulus and strength of each sample.

#### 2.3.3. Impact Test

Izod impact tests were carried out at ambient temperature according to the ASTM D256 standard by using a pendulum-type impact tester (Tinius Olsen Co., Model 892, Horsham, PA, USA). The dimensions of each specimen were 65 mm × 12.5 mm × 3 mm. Each specimen has a V-shaped notch, made by using a notch cutter according to the standard. The impact speed of the pendulum was 3.46 m/s. The impact distance was 610 mm. The impact energy was 12.66 J. The average impact strength of each sample was obtained from eight specimens.

#### 2.3.4. Heat Deflection Temperature Measurement

The heat deflection temperature (HDT) of each sample was measured with a three-point bending manner according to the ASTM D648 standard by using a heat deflection temperature tester (Model 603, Tinius Olsen Co., Horsham, PA, USA) equipped with a silicone oil bath. The specimen dimensions were 125 mm × 12.5 mm × 3 mm. The heating rate was 2 °C/min. Each HDT value was obtained at the deflection of 0.254 mm under the load of 0.455 MPa. The average HDT of each sample was obtained from three specimens.

## 3. Results and Discussion

### 3.1. Fracture Surfaces of Kenaf Fiber/PP Composites Made with Virgin PP and Waste EPP

Figure 3 displays SEM images of the fracture surfaces of kenaf fiber/PP composites produced using virgin PP with varying the kenaf fiber contents. The virgin PP sample exhibited a typical ductile fracture pattern, which can be found in PP. Meanwhile, the composites containing the kenaf fibers randomly distributed in the PP matrix, which was transformed from solid PP by melting during the injection process, exhibited many pull-out fibers and loosely placing fibers. It was observed that there were poor interfacial contacts and gaps, resulting in weak interfacial bonds between the hydrophilic kenaf fiber and the hydrophobic PP matrix. It can be explained by that kenaf fibers are lighter than virgin PP beads, hence, the virgin PP was fed somewhat faster than the chopped kenaf fibers into the hopper of the injection machine, causing some feed-timing lag. Accordingly, poor mixing between the kenaf fibers and the PP beads possibly occurred during the injection molding process. It may be said that the fracture surfaces were responsible for less effective enhancement of the mechanical properties of resulting composites, although the addition of natural fibers to the polymer matrix played a role in increasing the mechanical properties, as studied elsewhere [31,32,33].

Figure 4 shows the fracture surfaces of the PP matrix derived from the waste EPP showing a typical ductile pattern, as similarly observed from the virgin PP. In the case of the composites produced using waste EPP, the pull-out fibers were less protruded on the fracture surface, the kenaf fibers were more tightly surrounded by the matrix, and the fiber–matrix contacts were enhanced, compared to the composites with virgin PP. During the injection molding process, the chopped kenaf fibers were regularly fed into the hopper together with the crushed waste EPP without the feed-timing lag, and then they were compounded well. It was noted that the waste EPP of low specific gravity had much larger surface areas than the solid PP beads, giving rise to easy melting and mixing with the kenaf fibers fed under the same injection processing condition. As a result, the crushed EPP waste was completely melted and uniformly mingled with the chopped kenaf fibers, being compacted between the gaps of kenaf fibers.

Accordingly, it was obvious that the fiber–resin mixing in the kenaf/PP composite produced using the crushed waste EPP was better than that with the virgin PP, resulting in the lowered resistance to the applied mechanical load. The fracture surfaces were consistent with the mechanical and thermal results described in the following.

### 3.2. Effect of Waste EPP on the Flexural Properties

When the composites are subject to flexural environments, the stiffness or rigidity of composites is of great importance with their use in engineering and structural applications. Therefore, the flexural properties of a composite material are often of interest. Flexural modulus and strength are the ability of the material to withstand bending forces applied perpendicular to its longitudinal axis. The stresses induced by the flexural load are a combination of compressive and tensile stresses [34].

Figure 5 compares the flexural modulus and strength of kenaf fiber/PP composites produced using virgin PP and waste EPP. As described earlier, the virgin PP and the PP used to produce commercial EPP were identical. The difference between the virgin PP and waste EPP was the form of the material. The virgin PP was of bead form, whereas waste EPP was crushed EPP. As a result, the flexural properties between the virgin PP and waste EPP were comparable each other. The properties of the crushed EPP were even slightly higher than the virgin EPP due to better melting and mixing during the injection process. The flexural moduli of the samples (0 wt.% kenaf fiber) made with virgin PP and waste PP were 619 and 649 MPa, respectively. The flexural modulus of the PP sample prepared with waste EPP was slightly higher than that prepared with virgin PP. The modulus of the composite was gradually increased with increasing the kenaf fiber content, indicating remarkable property enhancement of about 65% with virgin PP and 210% with waste EPP at 30 wt.%, respectively. The flexural strengths of the samples made with virgin PP and waste PP were similar, showing the values of 14 and 15 MPa, respectively. As with the modulus tendency, the strength was gradually increased with increasing the kenaf fiber, indicating considerable property enhancement of about 43% with virgin PP and 106% with waste EPP at the 30 wt.% kenaf fiber contents.

Both the flexural modulus and strength of the composites produced using waste EPP were markedly higher than those produced using virgin PP, indicating the significant reinforcing effect. The flexural modulus and strength of the composites with waste EPP were significantly higher than those of the composites with virgin PP: 20% and 25% at 10 wt.% kenaf; 63% and 47% at 20 wt.% kenaf; 98% and 55% at 30 wt.% kenaf fiber contents. This was ascribed to better fiber distribution in the matrix and to the increased interfacial contacts between the fiber and the matrix in the composites produced using waste EPP. In addition, the fiber pull-out behavior, voids, and microstructural defects, which resulted from the poor fiber–resin mixing, were also responsible for the lowered mechanical properties of the composites with virgin PP, compared to the composites with waste EPP. It was noticed that the extent of the flexural property enhancement of the composites produced in the present work was relatively higher than that of the composites produced using surface-treated kenaf fibers and commercial PP pellets prepared by both extrusion and injection processes at the corresponding fiber contents, as reported earlier [8,34].

### 3.3. Effect of Waste EPP on the Impact Strength

Figure 6 compares the Izod impact strengths of kenaf fiber/PP composites produced using virgin PP and waste EPP. The impact strengths of both composites were similar, showing the values of 33 and 34 J/m, respectively. The impact strength was gradually decreased with increasing the kenaf fiber contents, unlikely with the flexural result. The deceasing tendency of the impact strength by adding brittle natural fibers including kenaf to a ductile thermoplastic matrix was often found in other natural fiber composites, as reported earlier [35,36].

The composites produced using waste EPP exhibited the impact resistance higher than those produced using virgin PP: 26% at 10 wt.% kenaf; 30% at 20 wt.% kenaf; 31% at 30 wt.% kenaf. It may be explained by that the composites having the fibers tightly contacted with the PP matrix derived from waste EPP were able to more efficiently transfer the external impact energy to the neighboring fibers through the matrix than the composite having the fibers, more or less, loosely contacted with the virgin PP matrix. In addition, the fiber–matrix clusters with some voids in the composite with virgin PP may act as weakening points against the impact energy, resulting in less energy absorption.

As described above, both virgin PP and waste EPP exhibited the ductile fracture pattern. Kenaf fiber is quite brittle, compared to virgin PP and EPP waste. Obviously, the incorporation of brittle fibers into the ductile thermoplastic matrix resulted in lowering the impact strength. It was suggested that the impact resistance as well as the mechanical properties may be further enhanced by the surface treatment of natural fiber, which was not conducted in the present work, leading to an increase of the interfacial bonding between the fiber and the matrix, as widely studied with natural fiber composites elsewhere [34,37,38,39].

### 3.4. Effect of Waste EPP on the Heat Deflection Temperature

Figure 7 compares the HDT of kenaf fiber/PP composites produced using virgin PP and EPP waste. The HDT values without kenaf fiber (0 wt.%) were 64 °C with virgin PP and 66 °C with EPP waste. In both cases, the HDT was gradually increased with increasing the kenaf fiber contents due to the reinforcing effect, as similarly found in other fiber/polymer composites [29,30,40]. The increasing tendency of the HDT was similar with that of the flexural properties, as above-mentioned. The HDT result indicated that with increasing the kenaf fiber contents the composites were increasingly resistant to the 3-point flexural loading applied to the composite specimens exposed to elevating temperature environment in a silicone oil bath. At 30 wt.% kenaf fiber, the HDT (113 °C) of the composite with waste EPP was about 12% higher than that (101 °C) of the composite with virgin PP.

Such the increase of the HDT in the composites with EPP waste indicated that the reinforcing fibers acting along the through-thickness direction of the specimen under the HDT testing condition played an effective role in increasing the HDT. Good mixing and tight interfacial contacts between the kenaf fiber and the melted waste EPP also contributed to the enhancement of the mechanical and thermal properties in the resulting kenaf fiber/PP composites.

## 4. Conclusions

The feasibility of using waste EPP as recycled matrix for kenaf fiber/PP composites produced only by using injection molding process was explored, focusing on the effect of waste EPP on the property enhancement of resulting composites.

The flexural properties, impact strength, and heat deflection temperature of kenaf fiber/PP composites with waste EPP were markedly higher than those with virgin PP. The property enhancement was in the range of 20–98% for the flexural modulus, 25–55% for the flexural strength, 26–31% for the impact strength, and 7–12% for the HDT, depending on the kenaf fiber contents. The fracture surfaces indicated the good mixing of chopped kenaf fibers and crushed waste EPP during injection molding process, showing the enhanced fiber–matrix contacts at the interfaces and less fiber pull-out behavior. The present study emphasizes that costless industrial waste EPP has some potentials as recycled PP matrix for replacing the virgin PP matrix widely used in conventional natural fiber composites in the optic of the circular economy.

## Figures and Tables

**Figure 1 polymers-12-02578-f001:**
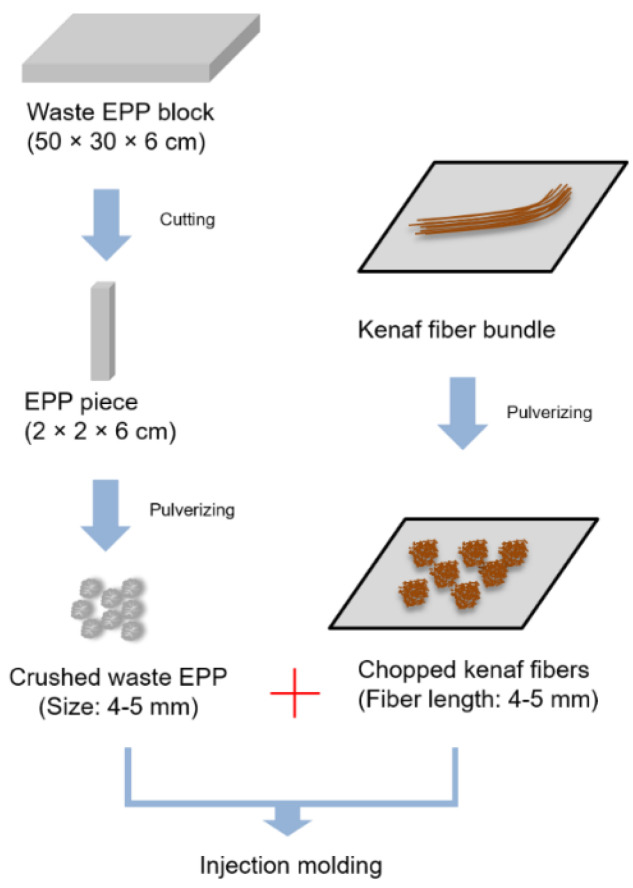
Crushed waste expanded polypropylene and chopped kenaf fibers obtained by pulverizing waste EPP blocks and kenaf fiber bundles, respectively, prior to injection molding.

**Figure 2 polymers-12-02578-f002:**
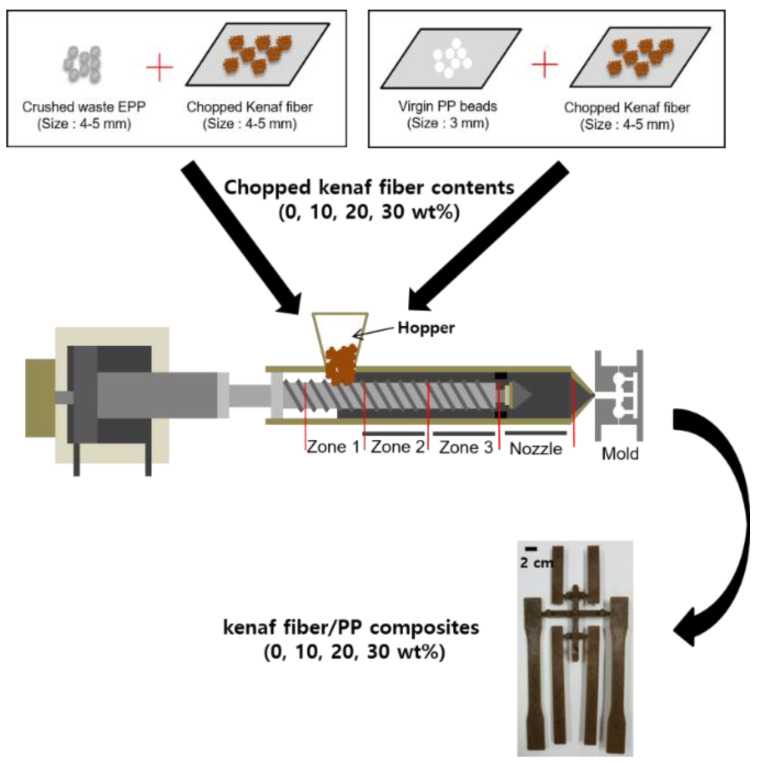
Injection molding process for producing kenaf fiber/polypropylene (PP) composites produced using waste EPP and virgin PP beads with different kenaf fiber contents, respectively.

**Figure 3 polymers-12-02578-f003:**
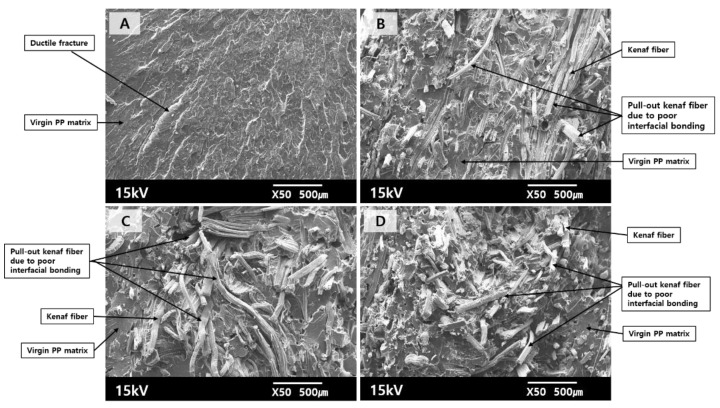
SEM images (×50) of the fracture surfaces of kenaf fiber/PP composites produced using virgin PP beads with various kenaf fiber contents ((**A**): 0, (**B**): 10, (**C**): 20, (**D**): 30 wt.%).

**Figure 4 polymers-12-02578-f004:**
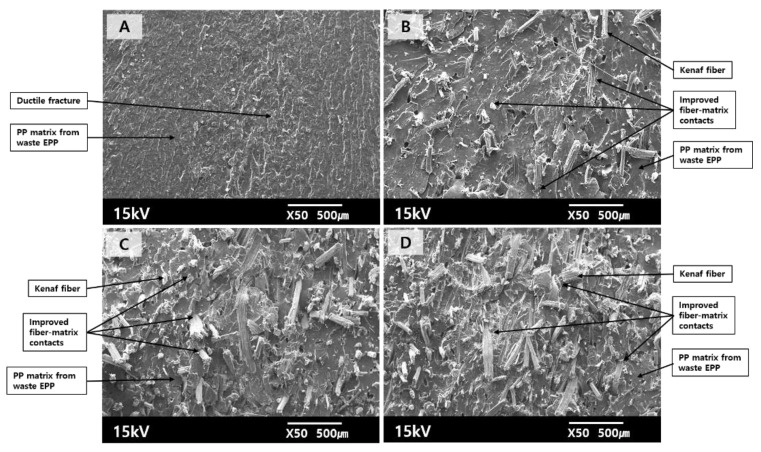
SEM images (×50) of the fracture surfaces of kenaf fiber/PP composites produced using waste EPP with various kenaf fiber contents ((**A**): 0, (**B**): 10, (**C**): 20, (**D**): 30 wt.%).

**Figure 5 polymers-12-02578-f005:**
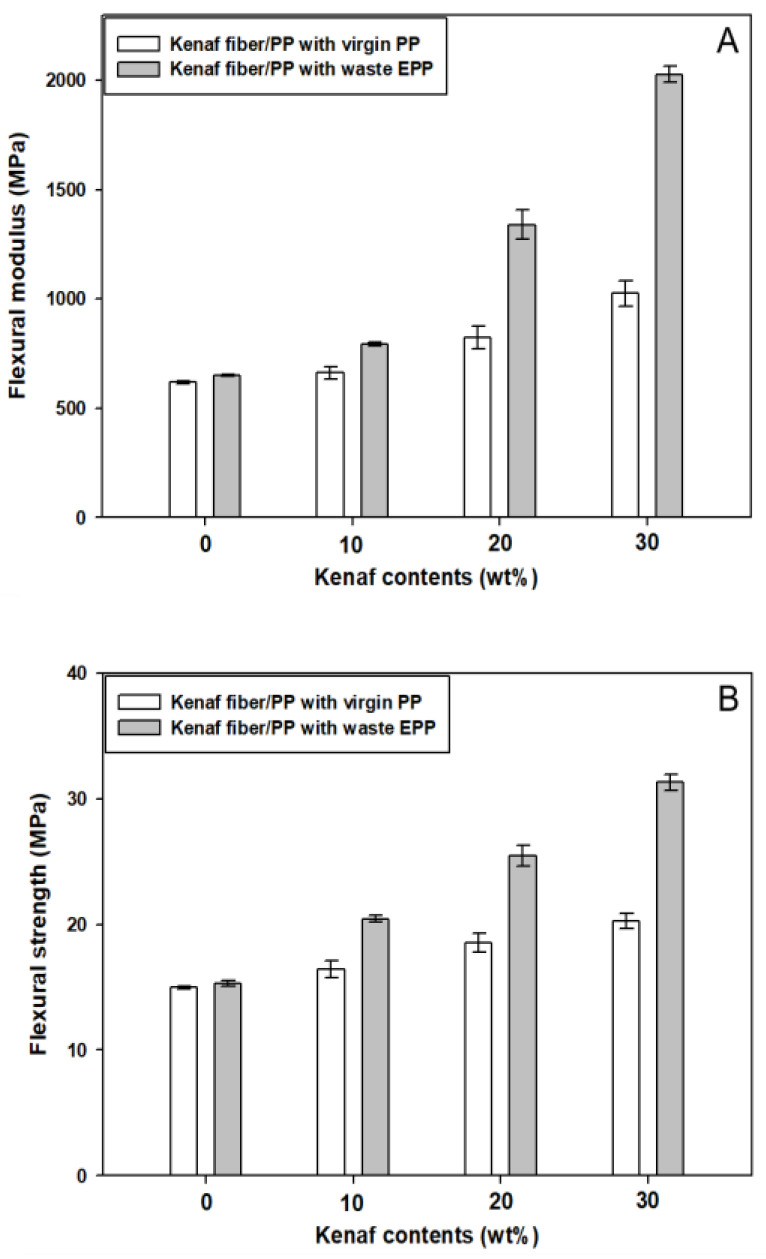
Flexural modulus (**A**) and strength (**B**) of kenaf fiber/PP composites produced using virgin PP and waste EPP with various kenaf fiber contents.

**Figure 6 polymers-12-02578-f006:**
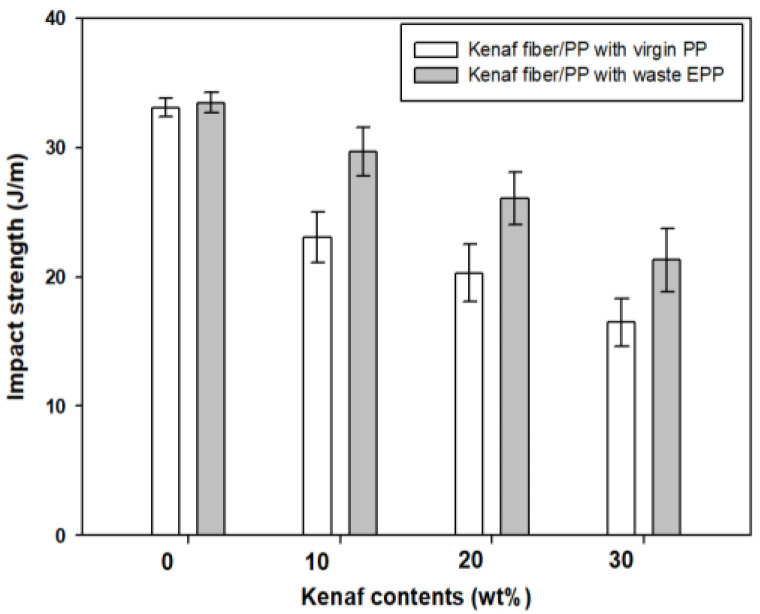
Izod impact strength of kenaf fiber/PP composites produced using virgin PP and waste EPP with various kenaf fiber contents.

**Figure 7 polymers-12-02578-f007:**
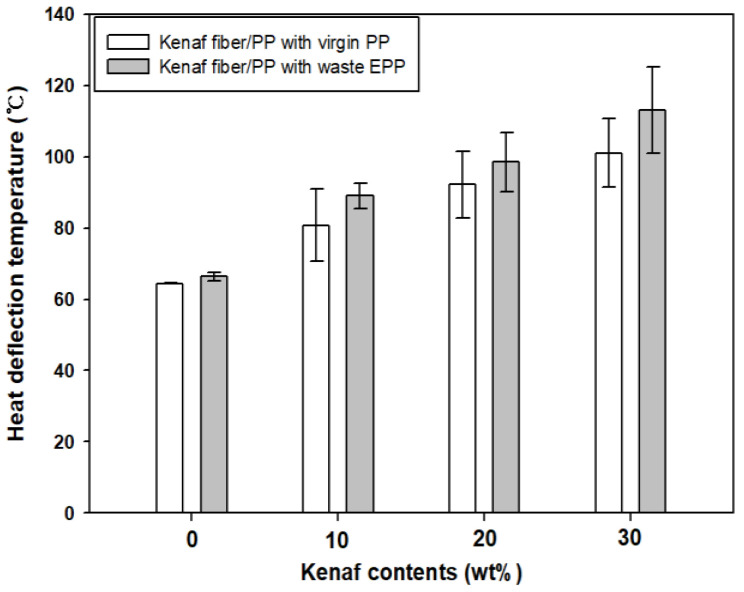
Heat deflection temperatures of kenaf fiber/PP composites produced using virgin PP and waste EPP with various kenaf fiber contents.

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
