# Peer review of "Effects of Waste Expanded Polypropylene as Recycled Matrix on the Flexural, Impact, and Heat Deflection Temperature Properties of Kenaf Fiber/Polypropylene Composites"

_polymers, 2020, doi:10.3390/polym12112578_

Round 1

Reviewer 1 Report

The paper entitled "Effects of Waste Expanded Polypropylene as Recycled Matrix on the Flexural, Impact, and Heat Deflection Temperature Properties of Kenaf Fiber/Polypropylene Composites" concerns the possibility to adopt recycled polypropylene for preparing kenaf fiber-reinforced composites. The mechanical performance of the composites strongly depends on the amount of kenaf addend into the polymeric matrix, also, the kind of polypropylene, virgin or recycled, adopted. Although interesting in the optic of the use of recycled polyolefins with improved mechanical performance, the paper lacks significant methodological features. 

  1. In the last years, many authors showed methods for the production of kenaf-PP composited by injection molding. The state of the art has to be improved (doi:10.1007/s12221-020-9600-x; 10.3390/polym11101707; 10.1088/2053-1591/ab2fbd)
  2. The mixing degree would depend on many factors: the rpm adopted in the screw, the residence time, etc. Certainly, the authors could improve the mixing, also in the injection molding machine. However, the method section lacks this important information (see https://doi.org/10.1177/0892705712461511).
  3. line 216-219. In the opinion of this referee, this finding is strange. The recycled materials are generally characterized by poorer mechanical performance than the virgin materials, thus the fact that the same amount of Kenaf induces better properties in the EPP instead in the virgin PP is due to the mixing conditions. Furthermore, the grade of the PP has to have a certain influence on the mixing conditions (due to a different viscosity, for instance) thus, one has to characterize, at least, the Mw of the recycled material (by GPC) in order to adopt a virgin PP with similar Mw for the comparisons.
  4. line 276-277. Interestingly, the virgin PP and the EPP exhibit similar values of the flexural modulus, even if the two materials show different behavior when the kenaf is added. This is a further indication that the authors should improve the mixing stage. Probably, the final (positive) finding will be that the recycle has a negligible effect on the mechanical performance, thus, the recycled PP could replace the virgin PP in the optic of the circular economy.

Author Response

Reply to the Reviewer’s Comments

Manuscript ID: polymers-974933

Title: Effects of Waste Expanded Polypropylene as Recycled Matrix on the Flexural, Impact, and Heat Deflection Temperature Properties of Kenaf Fiber/Polypropylene Composites

Comments and Suggestions for Authors

Reviewer 1

The paper entitled "Effects of Waste Expanded Polypropylene as Recycled Matrix on the Flexural, Impact, and Heat Deflection Temperature Properties of Kenaf Fiber/Polypropylene Composites" concerns the possibility to adopt recycled polypropylene for preparing kenaf fiber-reinforced composites. The mechanical performance of the composites strongly depends on the amount of kenaf addend into the polymeric matrix, also, the kind of polypropylene, virgin or recycled, adopted. Although interesting in the optic of the use of recycled polyolefins with improved mechanical performance, the paper lacks significant methodological features.

Comment 1. In the last years, many authors showed methods for the production of kenaf-PP composited by injection molding. The state of the art has to be improved (doi:10.1007/s12221-020-9600-x; 10.3390/polym11101707; 10.1088/2053-1591/ab2fbd)

Answer: According to the reviewer’s comment, the authors revised the manuscript with the state of the art improved by adding four references recommended. The revisions (lines 45-56) were marked in red in the revised manuscript.

Comment 2. The mixing degree would depend on many factors: the rpm adopted in the screw, the residence time, etc. Certainly, the authors could improve the mixing, also in the injection molding machine. However, the method section lacks this important information (see https://doi.org/10.1177/0892705712461511).

Answer: In the present work, the authors did not use extrusion technique, but only used injection molding machine. The main factors for injection molding were additionally given in the revised manuscript. There was a mistake to describe a word of the Materials and Method in the initial manuscript. The sentence “Kenaf fiber/PP composites were produced only by using injection molding process in the presence of extrusion process” in the initial manuscript was corrected to “Kenaf fiber/PP composites were produced only by using injection molding process in the absence of extrusion process” in the revised manuscript (line 107 in the revised manuscript).

According to the reviewer’s comment, the author added information on injection molding process in the revised manuscript (lines 115-117).

Comment 3. line 216-219. In the opinion of this referee, this finding is strange. The recycled materials are generally characterized by poorer mechanical performance than the virgin materials, thus the fact that the same amount of Kenaf induces better properties in the EPP instead in the virgin PP is due to the mixing conditions. Furthermore, the grade of the PP has to have a certain influence on the mixing conditions (due to a different viscosity, for instance) thus, one has to characterize, at least, the Mw of the recycled material (by GPC) in order to adopt a virgin PP with similar Mw for the comparisons.

Answer: As a matter of fact, the virgin PP and the PP used to manufacture commercial EPP used in the present work were same. Accordingly, it was expected that their molecular weights were corresponding. As shown in the paper, the mechanical, impact strength, and HDT between the virgin PP and the waste EPP were comparable each other. Although EPP was crushed, the properties were even slightly higher than the virgin EPP due to better melting and mixing during the injection process. The difference between the virgin PP and the waste EPP was the form of the material when they were used in the present work. The virgin PP was of bead form, whereas the waste EPP was crushed EPP. Here, the word “waste’ does not mean the waste indicating waste plastics such as waste PP, waste PE, etc., which can be normally purchased from the waste plastics market. Here the word ‘waste EPP’ indicates the EPP obtained by crushing process. The author thought that the fiber-resin mixing in the kenaf/PP composite produced using the crushed EPP (more like powder) was better than that with the virgin PP bead (larger PP size). It can be said that the crushed EPP with smaller size may be more easily melted than the virgin PP bead with relatively larger size, during injection molding process.

The relevant words and sentences were added in the revised manuscript (lines 95-97 & 264-269) (marked in red).

Comment 4. line 276-277. Interestingly, the virgin PP and the EPP exhibit similar values of the flexural modulus, even if the two materials show different behavior when the kenaf is added. This is a further indication that the authors should improve the mixing stage. Probably, the final (positive) finding will be that the recycle has a negligible effect on the mechanical performance, thus, the recycled PP could replace the virgin PP in the optic of the circular economy.

Answer: Thank you for your kind comments. Studying on further increasing the properties by improving the mixing stage is now under investigation.

Reviewer 2 Report

The authors present an interesting paper on the properties of kenaf reinforced PP and EPP composites.

The paper has merit to be published in polymers but I would like to make some considerations.

Why the authors do not prepare composite pellets prior to mold injection. This can increase the fiber dispersion and a better volume fraction precission.

I encourage the authors to proofread the paper. While now can be fully understood, there are some details than can be improved:

Line 31-32 “…biocomposites with reinforced properties” intead of reinforced perhaps is better enhanced or improved.

Line 52 “…with using…” one of the words is unnecessary…

Line 181, the authors comment the reasons for a weak interface. They can add that PP is hydrophobic and kenaf hydrophilic, that difficults the creation of bonds.

Figure 3 and 4 can be improved.

  1. Please increase its size
  2. Be sure that the scale can be read
  3. Add indication (arrow plus text) to help the references in the text.

Author Response

Reply to the Reviewer’s Comments

Manuscript ID: polymers-974933

Title: Effects of Waste Expanded Polypropylene as Recycled Matrix on the Flexural, Impact, and Heat Deflection Temperature Properties of Kenaf Fiber/Polypropylene Composites

Comments and Suggestions for Authors

Reviewer 2

The authors present an interesting paper on the properties of kenaf reinforced PP and EPP composites.

The paper has merit to be published in polymers but I would like to make some considerations.

Comment 1: Why the authors do not prepare composite pellets prior to mold injection. This can increase the fiber dispersion and a better volume fraction precission.

Answer: It has been well known that thermoplastic resins have advantages such as no cure reaction, recyclability, fast processing time, and good fracture toughness over thermosetting resins, whereas they also have processing difficulties due to high melt viscosity and low resin flow, causing inefficient resin impregnation. For this reason, extrusion and injection molding processes have been most frequently used processing methods for producing fiber-reinforced thermoplastics with chopped fibers. However, these processes often result in shortening of reinforcing fiber length, to be less than a few hundred micrometers. As a result, the mechanical and impact properties of resulting composites are much lower than expected. It restricts their extensive applications. Uses of increased fiber loadings may make thermoplastic composite processing more difficult because shear forces occurring between the screws in the barrel are increased during melt compounding process, resulting in further fiber damages and shortening. In addition, performing injection molding by directly feeding the fiber and resin without an extrusion stage gives us benefits making composite processing simpler.

Accordingly, in the present study the authors intendedly did not prepare composite pellets prior to injection molding, as described above.

The relevant words and sentences were added in the revised manuscript (lines 68-80) (marked in red).

Comment 2: I encourage the authors to proofread the paper. While now can be fully understood, there are some details than can be improved:

Line 31-32 “…biocomposites with reinforced properties” intead of reinforced perhaps is better enhanced or improved.

Line 52 “…with using…” one of the words is unnecessary…

Line 181, the authors comment the reasons for a weak interface. They can add that PP is hydrophobic and kenaf hydrophilic, that difficults the creation of bonds.

Answer: The above comments were added in the revised manuscript.

Figure 3 and 4 can be improved.

1.Please increase its size

2.Be sure that the scale can be read

3.Add indication (arrow plus text) to help the references in the text.

Answer: Figures 3 & 4 were improved. The figure size was increased, the scale was clarified, and the arrows with text was added according to your comments in the revised manuscript.
